# A Nationwide Survey on Working Hours and Working Environment among Hospital Dentists in Japan

**DOI:** 10.3390/ijerph17239048

**Published:** 2020-12-04

**Authors:** Tomoko Kodama, Yusuke Ida, Hiroko Miura

**Affiliations:** 1Department of International Health and Collaboration, National Institute of Public Health, Wako 351-0197, Japan; 2Healthcare Executive Program, Graduate School of Medicine, the University of Tokyo, Tokyo 113-0033, Japan; yskida@m.u-tokyo.ac.jp; 3Division of Disease Control and Epidemiology, School of Dentistry, Health Sciences University of Hokkaido, Hokkaido 061-0293, Japan; hmiura@hoku-iryo-u.ac.jp

**Keywords:** sustainable workforce, hospital dentist, gender, working hours, work environment, task-shifting

## Abstract

Sustainable human resource is one of the main issues in healthcare delivery and the way hospital dentists work has a significant impact on oral and dental healthcare services. This study is the first large-scale nationwide survey aiming to investigate the working hours including the working environment among hospital dentists in Japan. A total of 2914 hospital dentists responded to self-administered questionnaires from general hospitals (GHs) and medical educational institutions (MEIs) across the country. Among full-time dentists, the younger generation (i.e., those in their 20s and 30s) of both male and female dentists working in GHs engage in over 40 h of in-hospital clinical practice per week, apart from their self-learning hours. In contrast, the middle-aged dentists (i.e., those in their 40s and 50s) at MEIs work for more than 50 h on average due to the added teaching and research responsibilities. In a multiple logistic regression model using “more than 60 h of work per week” as the dependent variable, higher ORs (Odds Ratios) were found in males (OR = 1.83, 95%CI 1.50–2.22), MEIs (OR = 1.92, 1.52–2.42), and individuals specializing in dental and oral surgery (OR = 1.85, 1.47–2.32). Task-shifting was requested by 22.6% of the respondents for preventive care and dental guidance. Only a few male dentists experienced taking a parental leave and the peak distribution of working hours was shorter for females working in GHs. The support for child-rearing in the work environment is still insufficient and a consensus on the involvement of male dentists in childcare is needed.

## 1. Introduction

Sustainable human resource is one of the main issues in healthcare delivery and the way dentists work also has a significant impact on oral healthcare services. Oral diseases are recognized to be among the most prevalent diseases worldwide, reducing the quality of life of those affected and causing serious health and economic burdens [1]. Particularly, oral health in older adults is often poor and common oral diseases such as caries, periodontal disease, denture-related malconditions, hyposalivation, and oral pre- and cancerous conditions may lead to tooth loss, pain, local and systemic infection, as well as impaired oral function [2]. In addition, the presence of oral health problems was associated with greater risks of being frail and developing frailty in older age; thus, the identification and management of poor oral health in older adults could be important in preventing frailty [3]. According to the recent national survey among community habitants in Japan, 46.1–58.4 per 1000 people receive treatment for oral diseases, which are the third most treated diseases following hypertension and diabetes mellitus [4]. Japan achieved universal healthcare coverage in 1961 and dental care has been publicly guaranteed. The older adults tend to have multiple complications and sometimes their dental care is too complex to be handled in a normal primary care facility. As a result, there is a growing expectation for secondary dental care in community hospitals [5]. Therefore, countries with aging populations need to respond to the required dental and oral health services as the disease structure changes [6].

In order to provide oral healthcare services effectively and sustainably, it is also necessary to assess the workload of the involved health professionals. Occupational stress in the dental profession has been reported previously and various sources of stress were raised in their workplaces, such as the years of work experience and the number of patients treated daily [7,8,9]. In particular, the factors associated with burnout in dentistry were younger age, male gender, student status, high job-strain, and number of working hours in the clinical degree programs [10,11]. Therefore, task-shifting is an important strategy to reduce the burden of dentists. In the US, there is a shortage of dentists; thus, a new workforce development initiative would be needed to transform dental education and practice. Additionally, a strategy involving physician assistants and advanced practice registered nurses should be explored for the long-term expansion of oral healthcare access [12]. In England, factors that determine the career progression of dentists and dental healthcare professionals should also be assessed in order to optimize a sustainable workforce in the future [13].

In this study, we aimed to investigate the working state of hospital dentists and to clarify the related factors by means of self-reported working hours. With this, the time spent for clinical practice may be revealed as well as time spent on self-learning, meetings, and conferences, which are necessary for medical professionals but are sometimes undercounted in terms of hours spent. We also surveyed the factors influencing job sustainability, such as the parenting environment, unwillingness to work outside of urban areas, and choices for their future career.

## 2. Materials and Methods

### 2.1. Design and Participants

A nationwide cross-sectional study was conducted among hospital dentists in Japan in November 2018. The self-administered questionnaires were distributed to approximately 6500 dentists in 1632 all general hospitals (GHs) and 21 medical educational institutions (MEIs) providing dental practice. To protect the privacy of the dentists, personal collection envelopes were distributed at the same time as the questionnaires, and the dentists sealed the questionnaires in the collection envelopes. The hospitals (GHs and MEIs) collected them and returned to the research team. Responses were kept anonymous.

#### 2.1.1. Sociodemographic Characteristics

The survey items of the questionnaire for hospital dentists included items related to: age, gender, type of services (hospital administrator, full-time/part-time employment), area of specialty, family or relatives living in the same household (if any), and spouse’s or partner’s occupation. Regarding the areas of specialty in dental practice, four classified specialties were used according to the Medical Law allowance, such as “general dentistry”, “dental and oral surgery”, “pediatric dentistry”, and “orthodontics”.

#### 2.1.2. Working Hours and Work Sharing

The self-recorded time study was used to grasp the actual working time and hours for individual dentists, by filling in the time spent on “in-hospital practice”, “home visit (including nursing home visit)”, “teaching”, “research/self-learning”, and “conference/meeting/management work” recorded at 30 min increments for the week from November 15 (Thursday) to 21 (Wednesday), 2018.

“In-hospital practice” includes time spent in clinical practice seeing patients (inpatients and outpatients at hospital) including case conferences in the ward. The “Home-visit” category includes the time spent outside the hospital seeing patients at home or in care facilities (including time for transportation), as well as health screening activities outside hospitals. The “teaching” category includes time spent teaching and preparing for the education of the residents, dental hygienists and administrative staff. “Research and self-learning” include time spent on experiments, surveys, writing paper, and self-learning (e.g., reading dental journals and articles) at hospital, among others. This also includes training (e.g., study groups, workshops and lectures). The “conference/meeting/management work” category includes meetings and administrative tasks not directly related to medical practice (participation in meetings and committees, the management of duties or human resources). The classification of work content was followed by another national survey for physicians in 2019 [14].

For part-time hospital dentists, they were asked to fill in all working hours if they worked in several different hospitals or clinics. If the hospital dentists answered two or more areas of specialty, only one area was chosen for analysis in the following order: dental and oral surgery, pediatric dentistry, orthodontics, and dentistry.

#### 2.1.3. Working Environment

These were the conditions on childcare and nursing care for the aged parents (whether dentists have taken maternity leave, childcare leave, or nursing care leave), desired working style and actual working style; and the annual paid leave days acquired in the previous year (2017).

#### 2.1.4. Preference of Future Career

Employed practice (hospital or teaching institutions), opening their own clinic for practice, administrative positions (including local and central government or institutions), research or teaching (universities and research institutes), and others were included. Willingness to work in rural areas (the rural area was defined as outside Tokyo wards area, ordinance-designated cities, and prefectural offices).

### 2.2. Data Analysis

The differences in sociodemographic characteristics between male and female hospital dentists were examined by chi-squared and Fisher’s exact test. The Mann–Whitney U test was performed to examine the difference of dentists’ working hours according to gender and workplace (GHs and MEIs), respectively. The data for age were categorized as their 20s, 30s, 40s, 50s, 60s and older.

To examine the difference of total working hours across age groups, a nonparametric Kruskal–Wallis equality-of-populations rank test was applied and Dunn’s pairwise comparison was used for the multiple comparison with Bonferroni’s correction. The frequencies of total working hours per week were illustrated, including whether the distribution differed by living with children according to gender and workplace. A logistic regression analysis was conducted to examine the associations between working over 60 h per week and factors such as gender, age (categorical), working style (full-time), workplace (GH/MEI), and specialty of practice. A multiple logistic regression model was applied to those factors with statistical significance (*p* < 0.05) in simple logistic regression analysis. Using the chi-squared test, the reason for unwillingness to work outside of urban areas and direction of future career were compared between the full-time and part-time dentists. The statistical software STATA SE/16 was used for analysis.

### 2.3. Ethical Approval

This study was approved by the National Institute of Health Sciences Research Ethics Review Board (NIPH-IBRA # 12205).

## 3. Results

The questionnaire was sent to 1632 hospitals and responses were received from 2914 dentists (dentists response rate = 53.7%) from 480 hospitals (hospital response rate = 29.4%).

### 3.1. Sociodemographics and General Characteristics of the Study Subjects

#### Sociodemographic Factors of Respondents

The respondents comprised 1903 males (67%), 943 females (30%) dentists and 68 missing the information on gender (3%). The mean age of male dentists was higher (41.7, SD = 12.0) than female (34.8, SD = 9.01, z = −8.556, *p* < 0.001) indicated by the Mann–Whitney test. Table 1 shows the sociodemographics and characteristics of the study subjects according to gender. Among the participants, the 30s age group was most frequent (35.1%) among all age groups. For female dentists, the majority were in their 20s or 30s (73%), which was concentrated in the younger generations compared to male dentists.

As for type of services, 75% of them were in full-time and 22% were in part-time employment. However, the proportions of full-time was higher (82.5%) in male than female dentists (63.7%, χ^2^ = 138.3, *p* < 0.001). For their area of specialty in practice, “Dentistry (including preservation and prosthesis)” (53.4%) and “dental and oral surgery” (46.5) were frequent. The proportion of “dental and oral surgery” was higher in males (50.9%) than that of females (36.3%, χ^2^ = 69.9, *p* < 0.001) and the proportion of ‘pediatric dentistry’ was higher in females (10.6%) than males (6.0%, χ^2^ = 19.3, *p* < 0.001).

Among the hospital dentists who responded, 33.8% of them lived with children and this proportion was higher in male dentists (41.6%) than female (18.5%, χ^2^ = 150.3, *p* < 0.001). For the responded male dentists, 68.7% answered that their spouses or partners worked in a medical field. On the other hand, 98% of their spouses or partners of female dentists were medical professionals and the highest proportion (51%) were dentists. Among female dentists, almost 70% of their spouses or partners worked in the medical field. The spouses or partners working full time were frequently found to be of females (73.5%) than those of males (24.0%). Almost half of the spouses or partners of male dentists were unemployed (41.6%).

Regarding the annual paid leave acquisition rate, 36.7% did not take any off days at all while 36.5% took 1–5 days off. The frequency of night shifts (1–5 days) in the last 4 weeks was more frequent for males (14.2%) than females (8.4%, χ^2^ = 19.4, *p* < 0.001). Additionally, males were also more frequently on call (χ^2^ = 13.6, *p* < 0.001). Overall, 634 dentists comprising 21.8% of total respondents, were engaged in night shift or on-call duties. About half of the hospital dentists were willing to work in rural areas (40.5%). By age group, both male and female dentists in their 30s had the highest percentage (male 38.4%, female 39.0%, respectively).

### 3.2. Working Hours of Hospital Dentists

#### 3.2.1. Working Hours of Hospital Dentists

Figure 1 shows the percentage of engaged duties for 24 h among hospital dentists by gender from 6 a.m. on 20 November (Tuesday) to 5 a.m. of 21 November (Wednesday) as an example of self-recorded time study. This illustrates that most of hospital dentists started working from 8–9 a.m. and 50% still worked until 7 p.m. This graph also shows that dentists start research or management task after 5 p.m., especially male dentists. Most of the duties of hospitals’ dentists were clinical practice in the hospital and the other duties were counted as research/self-learning (20.1%), meeting/management (13.3%), teaching (5.7%) and home-visit (1.7%) at maximum.

#### 3.2.2. Weekly Working Hours

The total working hours and the contents of the work were shown by gender, age groups and type of service (full time/part time) in Figure 2 according to the workplace (GHs and MEIs).

Among the full-time dentists, the mean value of the longest total working hours was identified among male dentists in their 40s (59.9 h, SD = 15.9) at MEIs. Their average teaching hours and research/self-learning hours were 9.3 h and 18.5 h, respectively, which were more than double the length of time for those at GHs in the same age group. On the other hand, the time spent in in-hospital clinical practice was 30.6 h on average, which was 6.8 h shorter than that at GHs. In contrast, both male and female dentists working full time at GHs in the younger age group (20s and 30s) spent more than 40 h in in-hospital clinical practice; therefore, the total working hours exceeded 50 h. This figure clearly shows that full-time dentists at GHs spent almost 40 h of clinical practice on average, while their total number of working hours was 52.2 h on average. There was also a number of dentists who engaged in home visit practice for approximately 7 h per week. The mean value of meeting and management work was approximately 3 to 4 h in the GHs for all age groups, but the length of time increased from 4 to 10 h with the increasing age for male dentists working at MEIs.

The proportion of part-time dentists at MEIs (31.7%) was higher than that at GHs (12.6%). Among part-time dentists, males in their 20s and 30s and females in their 20s had almost same working hours as full-time dentists. The difference of total working hours among dentists between GHs and medical education institutions were found only in full-time male dentists by Mann–Whitney test (z = −4.656, *p* < 0.001). The gender differences were also found in both full-time (z = −3.288, *p* = 0.0010) and part-time dentists (z = −6.956, *p* < 0.0001) (not shown in figure).

#### 3.2.3. Factors Associated with Working over 60 h per Week

Table 2 shows the results of the logistic regression analysis for the factors related to working over 60 h per week. A simple logistic regression analysis indicates that the male gender (OR = 1.76, 95% CI = 1.46–2.13), working at MEIs (OR = 1.92, 95% CI 1.52–2.42), and specializing in dental and oral surgery (OR = 1.85, 95% CI 1.47–2.32) were positively associated with long working hours. However, age over 60 and certain specialty areas in dentistry (i.e., preservation and prosthesis) were negatively related. In Model I and Model II, the associations were kept in forced models.

#### 3.2.4. Time Distribution of Working Hours by Gender and Workplace

Figure 3 shows the time distribution of the average weekly working hours depending on whether dentists live with their children. The peak working hours of the female dentists at GHs who were living with children was shorter (30–39 h) than those without children (40–49 h). For male dentists who were not living with children, the peak of the average weekly working hours was 40–49 h and there was no obvious difference according to the workplace. However, the peak hours were even longer (50–59 h) in male dentists living with children and working at MEIs.

#### 3.2.5. Task Shifting

There were questions on the percentage of their daily tasks other than clinical procedures in the hospital. The most expected task was “explanations to patients and families” (19.6%). Other tasks, such as “filling in medical records (description of medical care etc.)” (15.8%), “preventive measures/dental health guidance” (13.2%), and “medical office work (providing medical information by leaflets or documents” (10.3%) were chosen from multiple questions. Among those tasks, 22.6% of hospital dentists agreed to share “preventive care/dental health guidance”, which is supposed to be covered with the dental hygienist. Other tasks, such as “medical office work” (16.7%), “explanation to patients and families” (9.6%), and “preparing medical records” (8.6%) were also chosen to be shifted to other staff (not shown in tables).

### 3.3. Working Environment for Who Needs Childcare or Nursing Care of Their Aged Parents

In this survey, 35 males (19.6%) and 144 females (80.4%) actually took paternity and maternity leave among the 1126 dentists (934 males, 192 females) who had children. Among the full-time male dentists, 74% wanted to work the same way as they did before having children, and 88% wanted no change in their actual working style. However, 16% of the respondents wished to reduce their workload and 13% wished to shorten their working hours while raising children. The highest percentage of full-time female dentists (48%) wanted to reduce their working hours, followed by 34% who wanted no change, 29% who wanted to reduce the number of working days, and 26% who wanted to reduce their workload. Furthermore, 10% of full-time female dentists and 21% of part-time female dentists faced the choice to “leave or resign”.

As an effective measure for continuing to work during childcare, many male and female dentists requested “Establishment/enrichment of in-house childcare facilities”. This was followed by “taking childcare leave” for male dentists and “securing personnel to work in shifts when necessary” and “promoting shorter working hours” for female dentists.

Regarding the long-term care of their family, 8% (239) of dentists answered that they had experiences before, and only five males and one female actually took care leave. Of those who answered that there was no change in the actual working style during nursing care, 20% of males and 34% of females showed a divergence from the desired working style. Additionally, 4% of dentists took leave/resigned from work because they had to care for their parents. For the multiple questions about effective measures for continuing work during the long-term care of the family, “securing personnel who can change work when needed” and “promoting the acquisition of paid leave” were the first options selected, followed by the “reduction of overtime work” and “reducing working hours”.

### 3.4. Unwillingness to Work Outside of Urban Areas and Future Career

In total, 51% of hospital dentists indicated their intention to work outside the city. The most frequent reason for unwillingness to work outside of urban areas was “to continue their desired line of work” (Table 3).

Between full-time and part-time dentists, “being medical personnel” for full-time dentists and “worrying about the work environment” for part-time dentists were also common reasons (*p* < 0.05). The reasons for unwillingness to work outside of urban areas varied by generation, with the most common reason for those in their 20s being “to continue work as I desire” and the next most common reason being “anxiety about working environment”. The overall trend is the same as in all age groups, even among those in their 50s and older, but as a characteristic of the generation, there was a decrease in items related to “children’s educational environment” and an increase in items related to “care for parents.” In addition, 42% of dentists responded that they did not wish to work outside urban areas (not shown in table).

The first choice for a future career among dentists was “employed practice” for both full-time and part-time dentists. However, the option to “open own clinic for practice” was more frequent for part-time dentists (26.3%, *p* < 0.001). Finally, “research or teaching” was more often chosen among full-time dentists (25.7%, *p* < 0.001).

## 4. Discussion

### 4.1. Representativeness of the Subjects

This study is the first large-scale nationwide survey that aimed to investigate the working hours including the contents of work, and the working environment among hospital dentists in Japan. There were 9825 registered hospital dentists (full-time 7705, part-time 2120) across the country at the time of study [15], and the responding dentists accounted for approximately 30% of them. Responses were received from all over the country in an unbiased manner and the distribution of gender and age groups of respondents were comparable to the national statistics, therefore, the representativeness of the data was assured to some extent. One of the reasons for this low response rate might be due to the time-consuming questionnaire which involved self-reported time study. Failure of the busiest dentists to respond may have led to an underestimation of the survey results. Another reason would be that the questionnaires did not fully cover hospital dentists, especially part-time dentists, because the questionnaires were distributed by hospital administrators.

There are 28 medical educational institutions which were established in the School Education Act (Law No. 26 of 1947) in conjunction with providing medical or dental education. Japanese dentists started working at the youngest age of 24 years old after graduating from high school and completing 6 years of dental education. For this survey, 21 institutions were randomly selected avoiding geographical bias. In this study, those several hospitals affected by the earthquake and heavy rain disaster across Hokkaido, and in Mabi-cho and Kurashiki City in Okayama Prefecture were not included in the study.

With regard to the gender balance of hospital dentists in Japan, the proportion of males was high among all dentists, but in recent years, the proportion of young females has been high (46.2%) for those in their 20s compared to 14.1% in their 60s in 2018, similar to what is reported in other countries [16]. This might be attributed to the 1986 Act on Securing Equal Opportunity and Treatment of Males and Females in Employment (Equal Employment Opportunity Act) in Japan.

### 4.2. Working Hours among Hospital Dentists and Expected Task Shift

To prevent pressure from employers to report accurate work hours and to ensure that responses to the survey were not detrimental to the dentist, the completed surveys were mailed individually to the research team. We cannot deny the results of this study in underestimating the working hours for those extremely busy dentists who could not reply. Due to the outliers in data distribution, we performed a non-parametric test for examining the difference between age groups.

Ogawa and Nishimura have discussed long working hours, depression, and burnout in younger physicians [17,18], and Saijo pointed out the importance of job strain assessment and preventive measures for depression [19]. The working hours of dentists are not as prominent as those of physicians reported in Japan, but they tend to exceed in dental and oral surgery and among young dentists in our study. The association between burnout and work engagement was shown in a preliminary study among US dentists [20] and other studies have reported the high levels of stress, burnout, and low well-being on dentists working in the United Kingdom and Hong-Kong [21,22]. The factors associated with work stress among dentists and students were described in a systematic review; they included younger age, male gender, student status, high job-strain/working hours, those enrolled in clinical degree programs, and certain personality types [10]. In our study, long working hours are found among males in general and younger dentists of both male and female dentists at GHs. Although we did not implement a stress check for the respondents, our findings could predict the high risk of burnout and work stress in these dentists, if necessary measures are not taken.

### 4.3. Excess Working Hours over 60 h per Week

In the multiple logistic regression analysis, factors such as male gender, working in MEIs, and specializing in dental and oral surgery were associated with working for over 60 h a week. With regard to the workplace, the peak percentage of working for over 60 h in GHs was 34.5% for male dentists in their 30s and 30.9% for female dentists in their 20s. The percentage of full-time dentists whose working hours (dental practice and others) exceeded 60 h was highest for full-time male dentists in their 30s, at 36.2% (not shown in table). However, the working hours of female dentists tend to decrease in their 40s after peaking at 22.7% in the 30s and then once again increase in their 50s and 60s. There was a 12.9% overlap of general dentistry in “dental and oral surgery”.

### 4.4. Annual Leave among Hospital Dentists

According to Labor Standards Act, Article 39, by Japanese law, annual leave is to be granted for 10 days 6 months after the date of hire, and annually thereafter. According to the Ministry of Health, Labor and Welfare, the paid leave acquisition rate in Japan is 52.4% in 2019, which is the lowest level in the world [23]. In principle, annual paid leave (hereinafter referred to as annual leave) is supposed to be given at the time when workers request it, but in view of the current low acquisition rate, the revision of the Labor Standards Act has triggered an opportunity since April 2019. In all companies, it was necessary for “employees who are given annual paid leave of 10 days or more a year, have the user specify the time period for 5 days of the annual paid leave”. From April 2024, the upper limit for overtime work will finally be applied to the medical field. Regarding the upper limit regulation of overtime hours for physicians, the current policy is generally expected to be “960 h per year, 100 h per month”, but with some exceptions, the standard of “1860 h per year, 100 h per month” is established [24].

### 4.5. Working Environment and Facility Work Engagement

There are broadly two types of hospitals providing dental care services in Japan, one that provides dentistry as part of a GH and the other that provides dentistry independently in MEIs. In our study, the analysis was divided in two: facilities MEIs and GHs. We examined the difference in working hours between those two types, because when dental care services are provided as part of a general hospital, dentists are expected to be under the same work management as are the physicians. As a result, dentists working in hospitals had similar working hours as medical doctors in a previous study [25]. According to the “Dental Health and Medical Vision Proposal” suggested at the review meeting of the Ministry of Health, Labor and Welfare in Japan in December 2017, GHs need to have special medical facilities that cannot be handled by dental clinics in order to respond to patients who require more specialized skills, depending on the hospital base and scale. In addition, there is also a need for improving the quality of local dental care by conducting regular training for local dental care workers [16]. Since the role of GHs in communities will continue to be significant, the management of the working environment of dentists is important.

### 4.6. Specialty Area of Practice in General Hospitals

Analysis by the area of practice revealed that the working hours, in general, for surgical procedures and perioperative medical care in dental and oral surgery were long, especially among 20-year-old dental and oral surgeons (weekly average of 50.4 h). It was equivalent to obstetrics and gynecology (50.6 h: excluding night/on-call duties) and pediatrics (50.2 h: excluding night/on-call duties) in Japan [25]. Other studies have indicated that the management of cross-cover between two specialties, such as oral and maxillofacial surgery and ear, nose, and throat surgery, could successfully reduce the working hours of junior doctors [26], and the centralization of surgical services enabled such operations to be conducted during normal working hours [27]. Therefore, efforts are needed to prevent these dentists working long hours to the point of burning out in Japan.

In Japan, from April 2024, as part of the reform of the working style of physicians, overtime work for medical doctors has been reduced to 960 h [24], however, there is no explicit reference to oral surgeons regarding the adaptation to overtime work.

Regarding work sharing, 22.6% of the dentists answered that current work can be shared with regards to preventive measures/dental health guidance, which is supposed to be shared with dental hygienists. The dentists agree that a greater amount of their current work could be shared with dental hygienists in principle, but they did not prefer a reduction in their current activity [28]. There exist clear differences among dentists in the way they take on their role as dentists with regard to business orientation and their willingness to distribute dental tasks to dental team members [29]. The survey revealed that the dentists indicated having a very high appreciation of the contributions of dental hygienists to their practice within legally allowable dental hygiene duties [30]. They also pointed out that good dentist–hygienist communication resulted in better patient satisfaction [22]. In order to reduce the workload among hospital dentists, there needs to be a balance between dentists’ role and willingness to share duties with dental care team members.

### 4.7. Adequate Work-Balance for Dentists—Childcare, Nursing Care, and Division of Duties

In this study, 42.9% of female and 77.8% of male dentists (full and part time) answered that they did not make any change while rearing children as they did not want to change their work. In other words, this means that 57.1% of female dentists and 22.2% of male dentists wanted to make some changes to their working style.

Previous studies have already reported that the flexibility of working hours is important to maintain a work–life balance and career satisfaction among dentists [31]. In our study, regarding the female dentists working at hospitals, the peak working hours among part-time dentists was 10–30 h per week. The female dentists have already chosen a flexible way of working part-time to maintain their own work–life balance. Alternatively, it may be harder for female dentists living with children to work full-time in GH departments than at MEIs.

This study clarified that male dentists’ working styles did not change with having children; in fact, they tended to work longer. According to recent national statistics in Japan, it was reported that males and females do not equally share housework and childcare duties and that females do seven times the housework done by males. In total, 60% of full-time female employees do more than 80% of housework [32]. This is also reflected in the results of this study, in which 88% of full-time male dentists answered that they did not change their work style while raising children.

In this survey, of the 945 men and 192 women who had childcare experience, the proportion of those who took paternity and maternity leave was 35 (3.7%) and 144 (75%), respectively, despite the leave being defined by the Labor Standards Law. There were 12 males (1.7%) and 97 females (50.2%) who took childcare leave for one year after paternity/maternity leave (not shown in the table). A Japanese researcher pointed out that the low rate of taking paid leave and childcare leave (among men) in Japan has probably risen because taking time off in itself is seen as a negative signal in a society that is valued for “working hard” [33]. There was a 14% gap between their actual work style (did not change) and their desire to change in this survey. This means male dentists were not able to actually change the way of working during childcare while they were hoping for some kind of change in their work style, such as a reduction in work content or shorter working hours. This shows that men’s childcare participation in Japan requires not only the intention of the person but also a change in social consciousness, especially at workplaces.

Similarly, there is a small number of dentists, 20% of male and 34% of females (8% of the total, 239) who have not been able to change their way of working as they wished. It is expected that the work style reforms will be promoted in the future.

Another reason that dentists hardly take leave is that they are responsible for the care of patients as their dentist-in-charge. After the analysis of the subitems of the survey, it was found that 45% of dentists work in those systems, which means that one patient always sees the same dentist. However, 53% of dentists work in a “seeing a patient with multiple dentists” system, which means that the patient does not have a designated dentist and sees any of the attending dentists. Additionally, 61% of part-time female dentists worked in the latter system.

### 4.8. Location Preference—Urban and Rural Areas

The uneven distribution of dentists is a problem, but half of the dentists who are willing to work outside urban areas mainly belong to the younger generation, under 45 years old. In addition, the reason for not wanting to work outside urban areas has been clarified to some extent in this study. In particular, the most common reason regardless of generation was the desire to continue one’s current work. There were also economic reasons and concerns about children’s education. In Australia, it was suggested that rural oral health could be improved by developing the oral health capacity of non-dental care providers and establishing effective communication between rural primary dental care providers [34]. Rural areas need flexibility and resources to develop innovative solutions that meet their specific needs. In addition, in rural areas, prevention needs to be at the front line of oral healthcare with systematic approaches that cover health professions and health sectors [28]. In the remote areas of Japan, it is expected further oral health promotion with multiple occupations and an adequate referral system from GHs to local private clinics is expected.

### 4.9. Limitations of this Study

Since the time study was self-recorded, recall bias in recording time could not be avoided. However, the results were acceptable in terms of actual conditions based on discussions with people on site. Although the response rate was relatively low, the age and gender distribution of the respondents was similar to that of the hospital dentists, as reported in national statistics [29,30]. Although there is no denying the possibility that people who work long hours answer more emphatically, we obtained sufficient data on the working hours and work–life balance of hospital dentists.

## 5. Conclusions

Long working hours should be carefully monitored to prevent health problems due to overwork among young generations, especially at GHs and in the specialty of oral surgery. A working environment for supporting child-rearing is still insufficient at hospitals and consensus on male dentists being involved in childcare is needed at the workplace. It is expected that health policy makers consider the young dentists’ willingness to contribute to work in rural areas and manage a nationwide sustainable workforce in hospital dentistry.

## Figures and Tables

**Figure 1 ijerph-17-09048-f001:**
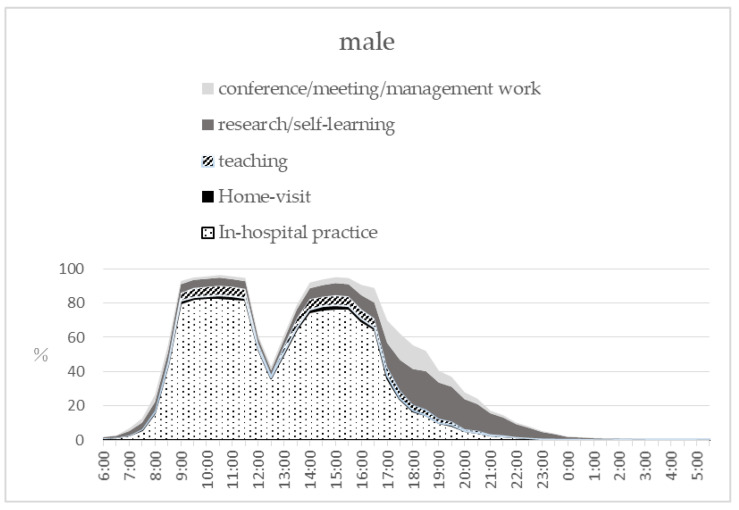
The percentage of engaged duties for 24 h among hospital dentists by gender.

**Figure 2 ijerph-17-09048-f002:**
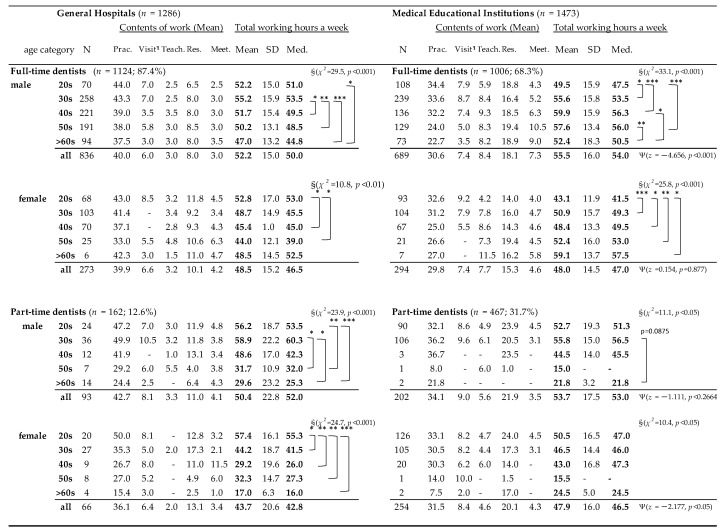
Weekly working hours among hospital dentists by gender, types of work (full time/part time), and age groups according to workplace. * *p* < 0.05, ** *p* < 0.01, *** *p* < 0.001. §KW test: Kruskal–Wallis equality-of-populations rank test was performed and Dunn’s pairwise comparison adjusted by Bonferroni’s method for multiple comparison: the number of hospital dentists who carried out home-visit practice was 332 (199 males and 127 females). Ψ: the difference in total weekly working hours between workplaces was examined by the Mann–Whitney test. Med: median, Prac: in-hospital practice, Visit: home-visit practice, Teach: teaching, Res: research or self-learning, Meet: meeting or management.

**Figure 3 ijerph-17-09048-f003:**
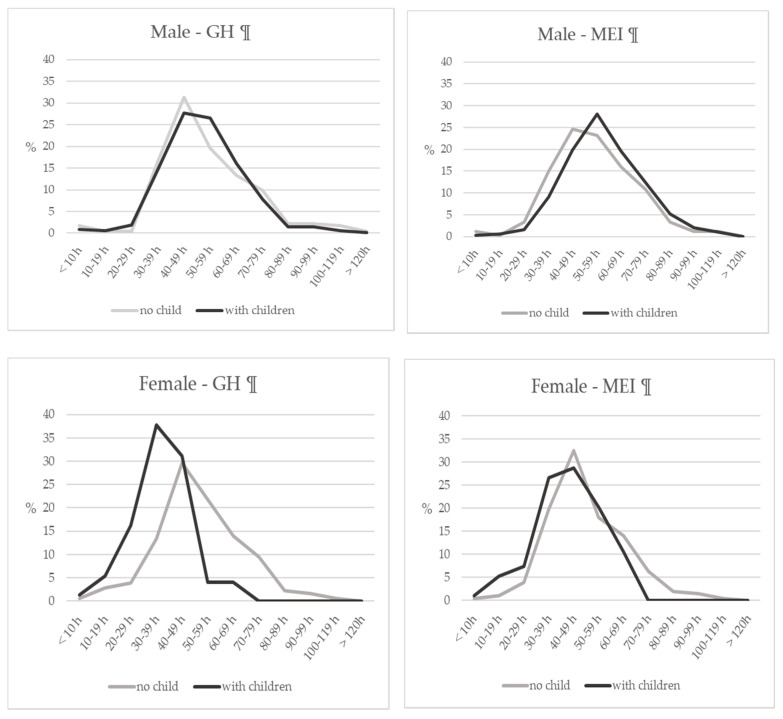
The time distribution of average weekly working hours depending on whether or not dentists live with their children. GH: general hospital, MEI: medical educational institutions.

**Table 1 ijerph-17-09048-t001:** Sociodemographics and characteristics of the hospital dentists who responded by gender (*n* = 2914).

Sociodemographhic Factors		Total		Male		Female	
		*n*	%	*n*	%	*n*	%
Age	20s ***	674	23.1	316	16.6	344	36.5
	30s	1023	35.1	658	34.6	351	37.2
	40s	563	19.3	383	20.1	167	17.7
	50s ***	409	14.0	341	17.9	57	6.0
	>60s ***	226	7.8	198	10.4	19	2.0
	missing	19	0.7	-	-	-	
Type of services							
Hospital administrator		36	1.2	30	1.6	4	0.4
Full-time employment ***		2186	75.0	1552	82.5	587	63.7
Part-time employment ***		641	22.0	299	15.9	331	35.9
Missing		51	1.8				
Area of specialty (duplicated)							
Dentistry (including preservation and prosthesis)		1555	53.4	1002	52.7	521	55.2
Orthodontics *		159	5.5	90	4.7	66	7.0
Pediatric dentistry ***		217	7.4	114	6.0	100	10.6
Dental and oral surgery ***		1354	46.5	968	50.9	333	35.3
Other *		221	7.6	128	6.7	88	9.3
Family or relatives living together (duplicate)							
Spouse or partner ***		1566	53.7	1222	64.2	305	32.3
Parents or relatives ***		429	14.7	201	10.6	221	23.4
Children ***		985	33.8	791	41.6	174	18.5
Others ***		52	1.8	22	1.2	29	3.1
No housemate ***		910	31.2	500	26.3	392	41.6
Missing		14	0.5				
Occupation of spouse or partner (*n* = 1606)				Male (*n* = 1245)		Female (*n* = 321)	
Dentist ***		512	32.8	335	26.9	163	50.8
Doctor ***		88	5.6	38	3.1	46	14.3
Dental hygienist ***		104	6.7	102	8.2	0	0.0
Dental technician		3	0.2	2	0.2	1	0.3
Other medical positions ***		185	11.8	175	14.1	9	2.8
Other ***		308	19.7	203	16.3	97	30.2
Unemployed ***		406	26.0	390	31.3	5	1.6
Type of services of spouse or partner (*n* = 1571)				Male (*n* = 1215)		Female (*n* = 317)	
Full time ***		536	35.0	292	24.0	233	73.5
Part time ***		404	26.4	353	29.1	42	13.2
Self-employed ***		104	6.8	64	5.3	37	11.7
Unemployed ***		527	34.4	506	41.6	5	1.6
Annual leave acquisition rate							
Not acquired		1070	36.7	661	36.4	382	43.5
1–5 days		1064	36.5	725	39.9	315	35.8
6–10 days		454	15.6	311	17.1	134	15.2
11–15 days		117	4.0	81	4.5	34	3.9
16 days or more		52	1.8	37	2.0	14	1.6
Missing		157	5.4	-		-	
Night shift for the last 4 weeks (days)							
None ***		2481	85.1	1622	85.2	859	91.1
1–5 ***		350	12.0	271	14.2	79	8.4
6–10		12	0.4	9	0.5	3	0.3
≥11		3	0.1	1	0.1	2	0.2
On call for the last 4 weeks (days)							
None ***		1850	86.5	1318	84.9	532	90.6
1–5 ***		277	12.9	226	14.6	51	8.7
6–10		9	0.4	7	0.5	2	0.3
≥11		3	0.1	1	0.1	2	0.3
Willingness to work in rural areas							
No		1475	50.6	762	40.0	419	44.4
Yes		1181	40.5	1006	52.9	469	49.7
20s *		390	32.1	199	19.8	181	38.6
30s		574	47.2	386	38.4	183	39.0
40s		282	23.2	194	19.3	81	17.3
50s *		175	14.4	151	15.0	18	3.8
>60s		83	6.8	76	7.6	6	1.3
NA		2	0.2	-		-	
Missing		258	8.9	135	7.1	55	5.8

* *p* < 0.05, *** *p* < 0.001 examined the difference between males and females by chi-squared test or Fisher’s test: the rural area is defined as outside Tokyo’s 23 wards, ordinance-designated cities, and prefectural offices NA: data not available SD: standard deviation.

**Table 2 ijerph-17-09048-t002:** Logistic regression analysis for the factors related to the working hours over 60 h per week.

Factors		Crude Model		Model I		Model II	
		OR	95% CI	(R^2^ = 0.0343)		(R^2^ = 0.0287)	
				OR	95% CI	OR	95% CI
Gender	Male	1.76 ***	(1.46–2.13)	1.95 ***	(1.60–2.38)	1.83 ***	(1.50–2.22)
Age	20s	ref		ref		ref	
	30s	1.29 *	(1.03–1.60)	1.29	(0.93–1.47)	1.25	(0.99–1.57)
	40s	1.07	(0.83–1.38)	0.96	(0.73–1.26)	1.03	(0.79–1.35)
	50s	1.08	(0.82–1.00)	0.96	(0.72–1.30)	1.00	(0.74–1.35)
	>60s	0.69	(0.48–1.00)	0.55 **	(0.37–0.82)	0.58 **	(0.39–0.86)
Type of services							
Full-time employment		1.09	(0.89–1.33)				
Workplace							
General hospitals		ref		ref		ref	
Medical educational institutions		1.22 *	(1.03–1.44)	1.50 ***	(1.25–1.80)	1.92 ***	(1.52–2.42)
Area of specialty							
Dentistry (including preservation and prosthesis)		0.58 ***	(0.49–0.69)	0.54 ***	(0.45–0.65)		
Orthodontics		0.81	(0.55–1.18)				
Pediatric dentistry		0.75	(0.54–1.05)				
Dental and oral surgery		1.32 **	(1.12–1.56)			1.85 ***	(1.47–2.32)
Family or relatives living together							
Spouse or partner		1.02	(0.87–1.20)				
Parents or relatives		0.94	(0.74–1.18)				
Children		1.03	(0.87–1.23)				
Others		0.71	(0.36–1.38)				
No housemate		1.08	(0.91–1.29)				
Spouse of partner’s job							
Full time		0.90	(0.73–1.12)			-	
Unemployed		1.04	(0.84–1.29)			-	

* *p* < 0.05, ** *p* < 0.01, *** *p* < 0.001. OR: Odds Ratio, CI: Confidence Interval

**Table 3 ijerph-17-09048-t003:** The location preference of work and the choice of future career among full-time and part-time dentists.

The Location Preference of Work and the Choice of Future Career	Full Time		Part Time	
	*n*	%	*n*	%
The first reason for unwillingness to work outside of urban areas	(*n* = 992)		(*n* = 275)	
To continue their desired line of work	608	61.3	154	56.0
For economic reasons (income/treatment)	51	5.1	20	7.3
Because there is no room for choice due to being medical personnel *	76	7.7	11	4.0
Because I am worried about the work environment *	57	5.7	27	9.8
Because the educational environment for children is not conducive	61	6.1	22	8.0
Because understanding from family is not obtained	62	6.3	17	6.2
For care of parents and relatives	37	3.7	5	1.8
Other *	40	4.0	19	6.9
The first choice for future career	(*n* = 2124)		(*n* = 630)	
Employed practice	1175	55.3	359	57.0
Open own clinic for practice ***	363	17.1	166	26.3
Administrative positions (including local and central government or institutions)	16	0.8	10	1.6
Research or teaching (universities and research institutes) ***	545	25.7	85	13.5
Others	25	1.2	10	1.6

* *p* < 0.05, *** *p* < 0.001 examined the difference between full-time and part-time dentists by chi-squared test.

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
