# Peer review of "A Nationwide Survey on Working Hours and Working Environment among Hospital Dentists in Japan"

_ijerph, 2020, doi:10.3390/ijerph17239048_

Round 1
Reviewer 1 Report
The authors report the results of an interesting survey on hospital dentists in Japan.
I only have some observations that could help the readers to clarify this large amount of data and one final proposal.
- Where the surveys sent to the Hospitals or directly to the dentists? In the first case, how did the Hospitals forwarded it to their dentists?
- Where the surveys anonymous?
- Is it correct that the 98% of female dentists husbands or partners were medical professionals? Which is the difference between the 98% medical professionals and the 70% workers in medical fields?
- Is the low percentage of teaching duties due to the fact that most of these Hospitals are not involved with Dental School Universities? If this is the case, do the authors think that it is not fair to compare dentists that have to teach with dentists that do not have this assignment? Please clarify.
It would be moreover very interesting to compare all these findings with data from the COVID era, in order to detect any interesting differences.
Author Response
Thank you very much for your review and comments.
We described the details of the survey in Material and Methods (L73-76)
.
-To protect the privacy of the dentists, personal collection envelopes were distributed at the same time as the questionnaires, and the dentists sealed the questionnaires in the collection envelopes. The hospitals (GH and MEI) collected them and returned to the research team. Responses were kept anonymous.
-It is correct that 98% of the husbands and partners of female dentists are in the health care professions. The 70% or more health care workers are on average for male and female dentists.
Therefore, we described the percentage among male dentists as well (68.7%) in L160-162.
-We believe that it is difficult to balance dental practice and teaching in general hospitals at present. Therefore, we think it is necessary to produce task-shifting and rebuild teaching system in the hospital as well. Yes, it would be valuable if we could compare all these findings in COVID era in the future and we appreciate your comment.
Reviewer 2 Report
The paper is an original study aiming to investigate the working state of hospital dentists in Japan, through the administration of questionnaires. The topic is interesting for multiple reasons: both for professionals, personnel managers and those responsible for organizing the health system.
The paper is well written and suitable for publication in the journal, after some minor revisions are made.
Introduction
Please explain briefly the healthcare system organization in Japan. In particular the dental care providing. Is it mainly public or private?
Materials and Methods
The design of the study is adequate. Data collection and analysis are well structured and the data are presented clearly.
Discussion
Were the involved hospitals evenly distributed on national territory? Or maybe there are some economic/social/ developmental differences between zones?
What is the typical working organization in hospitals? Is the dentist typically assisted by a nurse or dental assistant?
Author Response
Thank you very much for your review and comments.
The Japanese healthcare system is publicly guaranteed like medical care under universal health insurance (except for the use of special materials). Therefore, hospitals providing dental care are designed to be somewhat evenly distributed within the territory of the country. This is the reason for including the question of willingness to work in the rural area in this paper.
- We have added a brief note to the introduction. “Japan achieved universal health care coverage in 1961 and dental care has been publicly guaranteed.” (L45-46)
-As a typical working organization in hospitals, dentists are usually assisted by dental hygienists and dental assistants. However, in some hospitals, the number of dental hygienists is limited, which puts a strain on the dentists.
Reviewer 3 Report
The authors presented "A nation-wide survey on working hours and working 2 environment among hospital dentists in Japan".
Overall, its nicely presented, however there are few comments, which should be addressed.
The sample size calculation method should be mentioned.
The authors mentioned that the survey was conducted in November 2018, was it starting date or end date?
What was the respond rate from the teaching institutes?
Author Response
Thank you very much for your review and comments.
The sample size is as described in the discussion and the survey was launched sequentially on November 1st, and was deposited by November 30th. We were unable to calculate the separate response rates for GH and MEI.
Reviewer 4 Report
The paper entitled "A nationwide survey of working time and working environment among hospital dentists in Japan" is well written, clear and comprehensive. I don't think it needs further improvement. The authors performed a careful and detailed analysis and drew comprehensive results and conclusions.
Just two notes:
- could the authors report at what age dental career starts in Japan? (it's a detail, just curiosity to make the non-Japanese reader understand the average age at the beginning of the profession).
- The authors wrote "For female dentists, the majority were in their 20s or 30s (73%), which concentrated on the younger generations compared to male dentists." and "With regard to gender balance of hospital dentists in Japan, the proportion of males have been high among all dentists, but in recent years, the proportion of young females are high (46.2%) in 20s compared to 14.1% in their 60s in 2018, similar to what is reported in other countries [16]. Could you suggest a reason for these data?
Author Response
Thank you very much for your peer review and comments.
- Japanese dentists start working at the youngest age of 24 years old after graduating from high school and completing 6 years of dental education.(Added to L302-304 in discussion.)
- The number of dentists, as well as doctors and other professionals, has been gradually increasing, partly due to the Equal Employment Opportunity Act.
We added the sentence related to this in L 310-312.
“This might be attributed to the 1986 Act on Securing Equal Opportunity and Treatment of Males and Females in Employment (Equal Employment Opportunity Act) in Japan.”
Reviewer 5 Report
Thank you for inviting to review this manuscript. I have reviewed it carefully with a great deal of interest. The followings are my comments and suggestions to make the manuscript stronger for publication.
-Could you modify the alignment in the tables? In table 1 the alignment in the center makes it difficult to understand.
I think the left alignment could be better
-Table 3. Confidence intervals should be separated by commas. In case of negative values, hyphen overlaps the negative sign
-Please check the references, italicize the volume of the articles, and put the abbreviated jounal name.
(ACS style guide) Journal Articles:
Author 1, A.B.; Author 2, C.D. Title of the article. Abbreviated Journal Name Year, Volume, page range.
EXAMPLE:
- Saijo, Y.; Chiba, S.; Yoshioka, E.; Kawanishi, Y.; Nakagi, Y.; Itoh, T.; Sugioka, Y.; Kitaoka-Higashiguchi, K.; Yoshida, T. Effects of work burden, job strain and support on depressive symptoms and burnout among Japanese physicians. Int J Occup Med Environ Health 2014, 27, 980-992.
-Please check the website references. (REFERENCES 4, 5, 14, 15, 23, 24, 32)
Websites:
Title of Site. Available online: URL (accessed on Day Month Year).
Unlike published works, websites may change over time or disappear, so we encourage you create an archive of the cited website using a service such as WebCite. Archived websites should be cited using the link provided as follows:
Title of Site. URL (archived on Day Month Year).
Author Response
Thank you very much for your review and comments.
-As for alignment in the tables, we submitted the original table as you suggested, but the Journal has edited as you read according to their rule.
-As for the references, thank you for your suggestion about website. We used reference software for automatic editing and this process did not work well (lacked the information about website and accessed date etc). This time we corrected the references by hand but we are willing to utilize WebCite for future submission.